# A Capillary-Force-Driven, Single-Cell Transfer Method for Studying Rare Cells

**DOI:** 10.3390/bioengineering11060542

**Published:** 2024-05-24

**Authors:** Jacob Amontree, Kangfu Chen, Jose Varillas, Z. Hugh Fan

**Affiliations:** 1Interdisciplinary Microsystems Group (IMG), Department of Mechanical and Aerospace Engineering, University of Florida, Gainesville, FL 32611, USA; jma2260@columbia.edu; 2Department of Biomedical Engineering, Northwestern University, Chicago, IL 60611, USA; 3J. Crayton Pruitt Family Department of Biomedical Engineering, University of Florida, Gainesville, FL 32611, USA; varillas.perez.jose@gmail.com; 4Department of Chemistry, University of Florida, Gainesville, FL 32611, USA

**Keywords:** single-cell transfer, microfluidics, rare cells, circulating tumor cells

## Abstract

The characterization of individual cells within heterogeneous populations (e.g., rare tumor cells in healthy blood cells) has a great impact on biomedical research. To investigate the properties of these specific cells, such as genetic biomarkers and/or phenotypic characteristics, methods are often developed for isolating rare cells among a large number of background cells before studying their genetic makeup and others. Prior to using real-world samples, these methods are often evaluated and validated by spiking cells of interest (e.g., tumor cells) into a sample matrix (e.g., healthy blood) as model samples. However, spiking tumor cells at extremely low concentrations is challenging in a standard laboratory setting. People often circumvent the problem by diluting a solution of high-concentration cells, but the concentration becomes inaccurate after series dilution due to the fact that a cell suspension solution can be inhomogeneous, especially when the cell concentration is very low. We report on an alternative method for low-cost, accurate, and reproducible low-concentration cell spiking without the use of external pumping systems. By inducing a capillary force from sudden pressure drops, a small portion of the cellular membrane was aspirated into the reservoir tip, allowing for non-destructive single-cell transfer. We investigated the surface membrane tensions induced by cellular aspiration and studied a range of tip/tumor cell diameter combinations, ensuring that our method does not affect cell viability. In addition, we performed single-cell capture and transfer control experiments using human acute lymphoblastic leukemia cells (CCRF-CEM) to develop calibrated data for the general production of low-concentration samples. Finally, we performed affinity-based tumor cell isolation using this method to generate accurate concentrations ranging from 1 to 15 cells/mL.

## 1. Introduction

Single-cell analysis is critical for deciphering cellular heterogeneity, uncovering the unique roles and states of individual cells within complex tissues [1,2]. Rare cells such as circulating tumor cells (CTCs) are important targets for single-cell analysis [3,4,5]. By isolating and detecting these individual cells, researchers can delve into their unique genomic, transcriptomic, and proteomic profiles [6,7,8]. This elucidates their distinct roles in disease progression, development, or immune responses [9,10,11]. Microfluidics have been widely used for rare-cell isolation and single-cell analysis due to their precision and scalability [12,13]. Microfluidic systems, such as affinity-based [14,15,16,17], physical-property-based [18,19,20,21], and magnetic-ranking-based devices [22,23,24,25,26,27], have shown highly efficient rare-cell isolation. However, examining the performance of microfluidic devices for rare-cell isolation can be challenging due to the complexity of highly accurate sample preparation. For example, determining the efficiency of a microfluidic system for CTC isolation requires the preparation of spiked samples with tens of target cells in billions of peripheral blood cells. While serial dilution is commonly used to prepare spiked samples, the accuracy and reproducibility of this approach is questionable. For example, the recommended cell concentration for counting using a hemocytometer is around 10^6^ cells/mL. It requires five times of 1:10 dilution (additional times if a more accurate dilution ratio is used) to reach a scale of 10 cells per mL, leading to a large variation in sample preparation with accumulated dilution steps. These variations can ambiguate the performance of microfluidic devices. On the other hand, single-cell picking and transfer can tackle this dilemma through controlling the number of spiked cells with high precision and reproducibility [28,29]. Single-cell transfer (SCT) is a technique that relies on micropipettes to physically pick up an individual cell, and has evolved to numerous commercialized systems (sometimes called micromanipulators) such as CellTram 4r Air (Eppendorf, Hamburg, Germany) [30], DispenCell^TM^ (Molecular Device, San Jose, CA, USA) [31], CellenONE (Scienion, Berlin, Germany) [32], Biomek i-Series (Beckman Coulter, Brea, CA, USA) [33], etc. Microfluidics-based single-cell manipulation systems such as droplet-based cell sorting [34,35], microwells [36,37], optical tweezers [38,39], or microtraps [40] have also been reported. While widely explored, these costly platforms usually require robotic sample handlers with precise fluidic control, which limits the flexibility of the sampling area and increases the overall cost of sampling.

In this work, we report our study on the development of a capillary-force-driven SCT for accurate cell manipulation without the requirement of a sample handler. The SCT capillary tip can be easily fabricated with high reproducibility using temperature-controlled heat pulling. Mounted on an X-Y-Z stage, the SCT can locate target cells with high precision. The SCT can be switched to an ‘on’ or ‘off’ state by simply disconnecting or connecting the rare end of the capillary to a syringe. The sampling volume is accurately controlled by the pressure difference of the ‘on’ and ‘off’ stage of the rare end of the capillary. In addition, a quantitative model was used to determine the surface membrane tension resulting from cellular aspiration to ensure cell viability. Finally, we implemented this method to replace the serial dilution approach for tumor cell isolation in a microfluidic device. By creating reliable low-concentration cell samples, we observed an improvement in terms of reduced experimental time and systematic errors. Our results demonstrate that a more economical and simplistic method can be developed as an alternative to the conventional serial dilution approach or expensive commercial SCTs instruments without sacrificing accuracy or precision.

Table 1 lists some of the low-concentration transfer methods discussed above. Their accuracy, the number of components in the assembly, and price were compared. Our method retains the accuracy of other manipulators while reducing both the overall price and the number of components. More sophisticated, automated cell-picking systems such as CellCelector from ALS often cost more than USD 100,000.

## 2. Materials and Methods

Cell Culture. CCRF-CEM cells (human acute lymphoblastic leukemia) were purchased from American Type Culture Collection (ATCC, Manassas, VA, USA). Cells were cultured in RPMI 1640 supplemented with 2 mM glutamine, 10% fetal bovine serum (FBS; heat-activated; GIBCO), and 100 units/mL penicillin–streptomycin (Cellgro, Manassas, VA, USA) using incubation parameters of 37 °C under 5% CO_2_ atmosphere. These cells functioned as a surrogate of rare cells or CTCs.

Reagents and Buffers. The biotinylated aptamer Sgc8 [41] was used to isolate CCRF-CEM cells that were spiked in a buffer or blood. The sequence of Sgc8 is 5′-ATC TAA CTG CTG CGC CGC CGG GAA AAT ACT GTA CGG TTA GAT TTT TTT TTT-3′-biotin. Dulbecco’s phosphate-buffered saline (DPBS) without calcium and magnesium was used to wash cells. A buffer containing 10 mg/mL (1%) bovine serum albumin (BSA) in DPBS was used for rinsing the unbound molecules from the channel surface and resuspending cells in DPBS for cell capture. BSA in DPBS was used to passivate the surfaces to reduce nonspecific adsorption of cells in the channels. Anti-coagulant-containing whole blood from healthy participants was purchased from LifeSouth Community Blood Center (Gainesville, FL, USA) and used for all spiking experiments.

## 3. Results

### 3.1. Tumor Cell Micromanipulation

As shown in Figure 1a, our device for single-cell transfer consists of three linear stages (Newport, Irvine, CA, USA) mounted together to provide three-axis control. The X and Y axis model stages were joined together using 3D-printed, press-fit dowel pins made from acrylonitrile butadiene styrene (ABS). The Y and Z axis were joined using 90-degree angle brackets and flat speed nuts. A 3D-printed ABS tubing holder was mounted onto the Z-axis model 423 stage to house the syringe tubing (2.5 mm outside diameter, or O.D., 1.15 mm inside diameter, or I.D.). To create an interface between both the capillary and syringe tubing, a 10 μL pipette tip (Eppendorf, Hamburg, Germany) was cut to size and epoxied to the end of the tubing holder. After mounting the pipette tip to the holder, the tapered capillary was inserted into the pipette tip and Parafilm was used to seal any gaps between the bonds. This design inserts the capillary into the cell solution at an approximate angle of 45° with the horizontal to provide an increased field of view. To transfer rare cells to a new medium, such as a buffer or a blood sample, a tapered capillary is used to induce capillary action for cellular aspiration. The capillary was connected with the syringe tubing and the interface was perfectly sealed before use while holding the interior capillary pressure constant (at atmospheric pressure) using a 3 mL BD syringe (Becton Dickinson, Franklin Lakes, NJ, USA). As shown in Figure 1b, the capillary was lowered directly above the target tumor cell in preparation for capillary wetting. In Figure 1c, the capillary was inserted at a 45° angle into the cell solution to approach the target cell. At this step, only minimal amount of fluid was driven to the capillary by the capillary force as the pressure in the tubing increased when the air was being compressed. After the cell was correctly targeted, the tubing was manually disconnected from the syringe, bringing an abrupt pressure drop in the rear end of the capillary. Fluid rushed into the capillary driven by capillary force and aspirated a portion of the target cell’s outer membrane from the abrupt pressure change (Figure 1d). Next, the tubing was re-connected to the syringe thus retarding fluid withdrawal. Finally, the capillary was relocated to the target solution (e.g., a buffer) and the syringe plunger was slowly depressed until the fluid was released (Figure 1e). 

### 3.2. Tapered Capillary Optimization

The tapered capillaries start with an initial 1.5 mm O.D./1.12 mm I.D. borosilicate glass micropipette. Each micropipette was clamped into the two jaws of the capillary puller (Narishige PB-7, Narishige Group, Tokyo, Japan). For creating capillaries for SCT, the voltage applied to the heater was 1.75 V to create a gravity-driven taper from 1.12 mm to a final tip I.D. of 2.0 ± 0.1 µm (Figure 2a). The ending tip I.D. was measured for a given voltage range to determine an I.D. spread for the process (Figure 2b). The capillaries were cut 4 cm from the tip using a microforge (Narishige MF-900, Narishige Group, Tokyo, Japan) heated with 60% of maximum puller power to melt and slice accurately. All capillaries were inspected using an Olympus IX71 microscope (Olympus Corporation, Shinjuku City, Tokyo, Japan) to spot possible defects in fabrication. Before use, small batches of pulled capillaries were tested by inserting each capillary into deionized (DI) water to ensure the tip could withdraw fluid at the given cutting and sizing conditions. 

### 3.3. Determination of Equivalent Pressure Change

When a hydrophilic capillary was soaked in a fluid, the fluid surface inside the capillary would rise due to capillary action and finally reached an equilibrant position. We used equivalent pressure difference between the capillary and the atmosphere to describe the capillary action. The effective pressure was defined as the stationary pressure produced by the rising fluid inside the capillary. Using the Young–Laplace equation for pressure jump across a tube interface, the effective pressure was determined through Equation (1). For a water-filled glass meniscus in atmosphere conditions at sea level at room temperature, the surface tension is *γ* = 0.0728 J/m^2^ [42], contact angle is *β* = 32.3°, and *R* = 0.56 mm for the I.D. of the un-tapered section of the capillary. The effective pressure was calculated using Equation (1) to be 220.5 Pa. Jurin height, *h*, was theoretically and experimentally measured to ensure the approximation for *R* was accurate. Using Equation (2) and *ρ* = 997 kg/m^3^ for water and *g* = 9.81 m/s^2^, Jurin height was found to be 22.5 mm. By inserting the tapered capillary into dyed water, Jurin height was found to be 23.6 ± 0.1 mm.
(1)$$\Delta P=\frac{2\gamma cos\;(\beta )}{R}$$
(2)$$h=\frac{2\gamma cos\;(\beta )}{\rho gR}$$

The length of fluid occupying the capillary is thus calculated as
(3)$$L=\frac{h}{sin\;\theta }$$
where *θ* is the angle of insertion. In this work, *θ* equals to 45°.

Before cell sampling, the capillary tube with the rare end connected to a sealed 3 mL syringe was dipped in the solution to sample a small amount of fluid and build up the inner pressure of the SCT system. Before the small volume sampling, the inner pressure of the SCT was $P_{1}$, which equaled to the atmosphere pressure (~1.01 × 10^5^ Pa), and the volume of the air in the SCT was denoted as *V_1_*, which was approximately 3.1 mL. After the small sampling, the inner pressure was denoted as $P_{2}$, with the corresponding air volume of *V*_2_ (Appendix A). Using the idea gas equation state to describe the status change as follows
(4)$$P_{1}V_{1}=P_{2}V_{2}$$
the volume change of the air can be described as
(5)$$V_{1}=V_{2}+n(\pi R^{2})L_{2}$$
where r was the inner diameter of the original capillary. n was the modification coefficient of the capillary tip volume, which was estimated to be 1/3. $L_{2}$ was the fluid length during the small volume sampling. It equaled to
(6)$$L_{2}=h_{2}/sin\;\theta$$

The balance between the inner pressure after small sampling with the atmosphere can be described as follows
(7)$$P_{2}+\rho gh_{2}-\Delta P=P_{1}$$
where ΔP was the effective capillary driven pressure as calculated from Equation (1). Combining Equations (1)–(7), we can obtain a second-order equation regarding the single unknown variable $h_{2}$, as given below
(8)$$h_{2}^{2}-\left(\frac{\Delta P}{\rho g}+\frac{V_{1}sin\theta }{n\left(\pi R^{2}\right)}+\frac{P_{1}}{\rho g}\right)h_{2}+\frac{\Delta PV_{1}sin\theta }{\rho gn\left(\pi R^{2}\right)}=0$$

Solving this equation, we can obtain the value of $h_{2}$ as 8.9 mm. The corresponding fluid length $L_{2}$ is 12.6 mm.

After initial small sampling, the SCT system was used to locate target cells. Once the target cell was located, the rear end of the capillary was disconnected from the syringe, and more samples was aspirated to the SCT. The volume of the sampling fluid was calculated as
(9)$$\Delta V=(L-L_{1})\pi R^{2}$$

The resulting volume of fluid aspirated by the capillary was calculated as 21.6 µL. This precisely controlled sampling volume can be used to manipulate rare cells in suspension. The pressure difference between the ‘on’ and ‘off’ state was thus modified to be 133.3 Pa.

### 3.4. Analysis of Critical Membrane Tension Caused by Capillary Aspiration

Once the capillary was positioned near a tumor cell using the micromanipulator, the tubing would be disconnected from the capillary. A small suction pressure (~133.3 Pa, as calculated above) was produced via capillary action to attract the cell to the capillary tip. To ensure cell viability and proliferative potential were maintained after transfer, the experimental membrane tension was compared to the critical membrane tension for cell rupture. Using the calculated pressure drop, the maximum membrane tensions the cell experiences from aspiration was calculated through Equation (10).
(10)$$\tau =\frac{\Delta P}{2\left(\frac{1}{R_{CH}}-\frac{1}{R_{OUT}}\right)}$$

In Equation (10), ΔP is the change in pressure approximated through control experiments and both $R_{CH}$ and $R_{OUT}$ are the channel radius and the radius of cells used in this work, respectively. SCT uses the principle of micro-aspiration to entrap the desired tumor cells (Figure 3a). The maximum tensions at the surface of most clinical tumor cells before cell rupture occurs ranges from 800 to 5000 Pa/µm with a median of 3700 Pa/µm, using micropipette deformability studies [43]. Considering this upper limit, the membrane surface tension was quantified for varied tip I.D. and tumor cell diameters (Figure 3b). This demonstrates that under standard laboratory conditions, the aspiration-based capture induces membrane tension under the minimum threshold for cellular rupture.

### 3.5. Calibration of Single-Cell Transfer

To attain calibration data for this cell transfer method, the entire procedure was iterated for transferring 1, 5, 10, and 15 tumor cells. Immediately before cell isolation experiments, cells were harvested from the culture flask and resuspended at 10^3^ cells mL^−1^. The target CCRF-CEM cells were stained with Vybrant Dil (red) cell-labeling solution (Invitrogen, Carlsbad, CA, USA) at 5 µL/mL and then rinsed with DPBS without calcium and magnesium and resuspended at 10^3^ cells mL^−1^ in 1%BSA-containing DPBS. 

Once the tapered capillary was installed into the micromanipulator assembly and placed on the motorized microscope stage, 10 µL of DPBS was dropped onto a Hausser–Levy hemocytometer (Hausser Scientific, Horsham, PA, USA) to retain a spherical droplet due to hydrophobic surface modification. The syringe tubing was connected to a 3 mL BD syringe tube with the plunger at the 1 mL position. The capillary tip was inserted into the droplet for capillary wetting until the motion in the tubing halted. Next, 10 µL of the CCRF-CEM cell solution was deposited into the DPBS droplet. Using a scientific-grade CCD camera (Hamamatsu C4742-08-12AG) with the computerized monitoring system, the capillary tip was positioned within a 10 µm distance from the targeted CCRF-CEM cell. For SCT, the tubing was disconnected to produce a pressure drop and the resultant capillary force withdrawal allowed the tumor cell to delicately adhere to the tip inlet (Figure 4a). By simplifying the process down to a transfer from one medium to another (i.e., Figure 1b–e), we successfully generated a calibration curve for spiking low concentrations of tumor cells into a buffer (Figure 4b). The transfer efficiency of the SCT is up to 100% for small cell numbers, which is comparable with commercialized single-cell manipulation systems [30,31,32,33]. However, for higher cell numbers, the transfer efficiency of the SCT slightly decreases (86.7 ± 5.1% for 15 cells), possibly due to the fact that more time is required for transporting larger number of cells and cell loss is more likely.

### 3.6. Capture of Spiked Tumor Cells in Microfluidic Devices

Immediately before experiments, cells were washed and resuspended at 10^4^ cells mL^−1^. The target CCRF-CEM cells were stained with Vybrant Dil as described above. After centrifugation, the cells were resuspended in 100 µL DPBS. A total of 100 µL 500 nM DAPI was added to stain the CCRF-CEM cells for better visualization of the cell nucleus. After incubation for 25 min at 37 °C, the cells were washed with DPBS and resuspended at 10^3^ cells mL^−1^ in BSA-containing DPBS.

Anti-coagulant-containing human whole blood from healthy donors was purchased from Life South Community Blood Center and used for all spiking experiments. To reduce the effects of blood viscosity, 500 µL of whole blood was diluted with 500 µL of DPBS. To create a spiked solution at a desired tumor cell concentration, the SCT apparatus was used to transfer 1, 5, 10, or 15 tumor cells using the same method outlined in the calibration section. The contents of the capillary were deposited into the 1 mL blood sample by gently pressing the syringe plunger down until the cell solution was released. 

To functionalize the microfluidic device, 200 µL of 80% ethanol was used to wash the device, followed by a 200 µL washing step with DPBS. Next, 100 µL of 1 mg mL^−1^ avidin (Invitrogen) in DPBS was introduced into the device using a low-pressure vacuum pump, followed by incubation at room temperature for 15 min and then two washes with DPBS. Then, 100 µL of biotinylated aptamer Sgc8 was introduced into the device using a low-pressure vacuum pump, followed by incubation at room temperature for 15 min and then two additional washes with DPBS. Finally, 1 mL of the spiked blood sample was infused into the device at a flow rate of 1 µL s^−1^. A small magnetic bar was placed at the end of the syringe, being continuously stirred to prevent cell settling during infusion. After blood infusion, the channels were washed with 300 µL of DPBS at the same flow rate. The microfluidic device was placed on the motorized stage of an Olympus IX71 microscope and imaged using the CCD camera. The number of cells captured in the microchannels was determined by acquiring fluorescent images with the assistance of counting algorithms in the CellSens Standard (Olympus, Tokyo, Japan). Figure 4c shows a representative image of a CCRF-CEM captured on the aptamer-functionalized microchannel from a sample prepared using the SCT device. This method proved to be reliable for tumor cell concentrations ranging from 1 to 15 CTCs/mL of whole blood, as seen in the regression analysis presented in Figure 4d. With the correlation coefficient R^2^ being equal to 0.997, this SCT technique proves to be dependable at cell concentrations ranging from 1 to 15 cells/mL.

## 4. Discussion

We developed an alternative procedure for transferring a low number of cells for rare-cell studies. Our SCT system is able to perform capillary-driven, controllable aspiration micromanipulation of single cells without the requirement of pumping or sampling handler. We found that optimizing micropipette geometry and size allows for accurate cellular aspiration without the use of external pressure systems. 

Our SCT system was used to create a calibration curve to illustrate both its accuracy and its precision in CTC spiking at low concentrations. To further demonstrate its spiking abilities, we transferred CCRF-CEM cells to whole blood samples for CTC capture and enumeration using a microfluidic chip. Since the essential components of the SCT only include a tapered capillary, a syringe, and a micromanipulator, the system can be assembled in an economical manner. Although tumor cells were the focus of this report, our apparatus is expected to be applicable to other applications to replace traditional serial dilution procedures as well as commercial SCT methods. The current SCT platform focuses on single-cell transfer, which mainly targets floating cells. Therefore, it can easily target cell suspensions such as white blood cells. For epithelial cells and endothelial cells, they can be targeted and transferred if they are in a suspension state. However, if they are in an adhesion state, it is hard to aspirate them using the SCT without destroying them. For future studies, this system can be combined with adherent-cell-release technologies such as optical tweezers [44] to manipulate adherent cells.

## Figures and Tables

**Figure 1 bioengineering-11-00542-f001:**
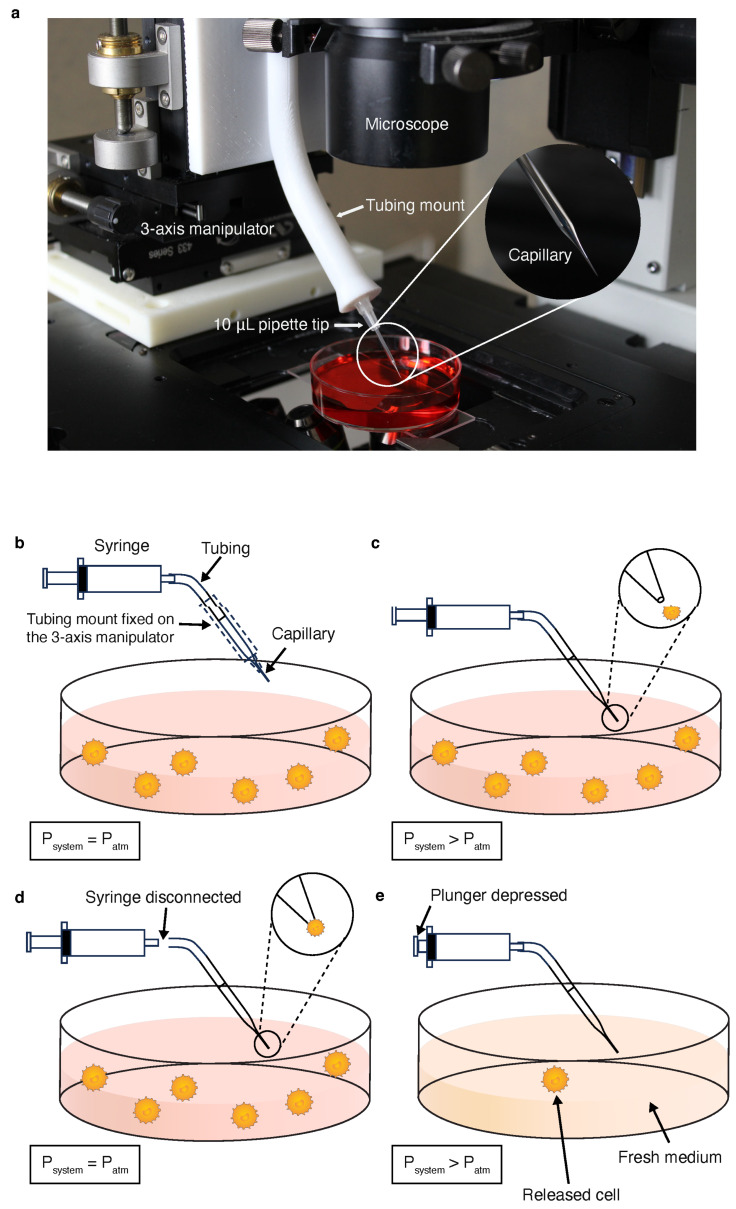
(**a**) Experimental apparatus of the SCT micromanipulator depicting its key features. Multiple axes of motion are created by combining three ball bearing linear stages into a single 3-axis stage. The micromanipulator is mounted on the microscope stage, allowing for both digital and manual monitoring during transfer. (**b**) Capillary-action-enabled cell capture set-up with pressure closed off to the atmosphere with a capillary positioned above a target cell. (**c**) Initial fluid withdrawal upon the capillary contacting the cell solution allowing the target cell to come into proximity with the tip. (**d**) Cellular-aspiration-based capture driven by capillary force after disconnecting the syringe tubing. (**e**) The aspirated cell is dispensed into a new medium by re-connecting it to the syringe tubing and depressing the syringe plunger.

**Figure 2 bioengineering-11-00542-f002:**
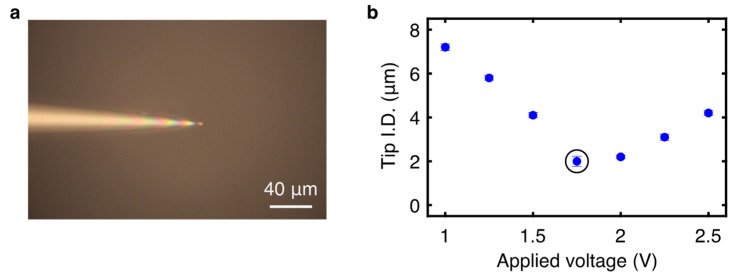
(**a**) Resulting SCT borosilicate glass capillary with 2 µm terminal I.D. (**b**) Capillary tip I.D. after pulling versus the voltage applied for capillary melting. The x axis indicates the voltage applied to the heating element; the y axis is the resulting tip I.D., with error bars indicating one standard deviation *(n* = 3). The data point within the circle indicates the optimal tip I.D. used for SCT.

**Figure 3 bioengineering-11-00542-f003:**
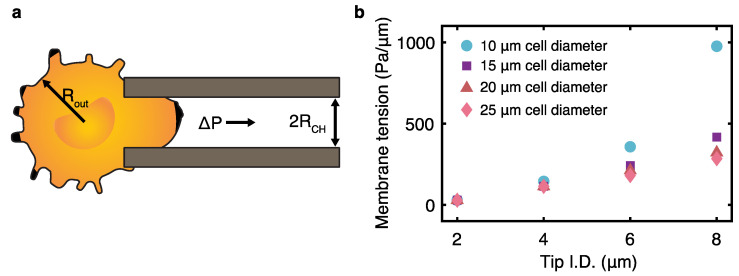
(**a**) Working principle behind cellular aspiration during SCT. (**b**) Cellular membrane tension as a function of capillary tip I.D. Multiple data series are shown for varied cell size. Increasing channel size results in a greater projection length into the capillary, thus resulting in higher shear stresses.

**Figure 4 bioengineering-11-00542-f004:**
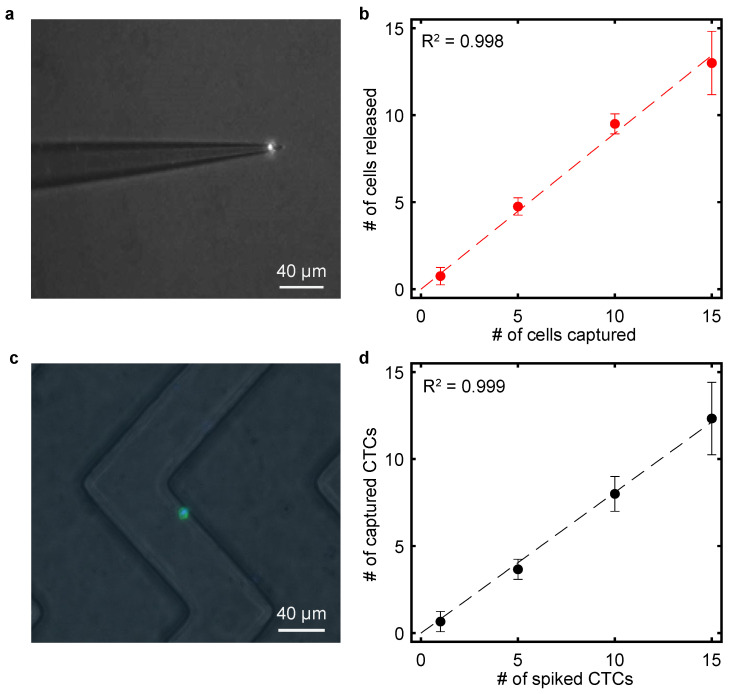
(**a**) SCT capture from a DPBS buffer using a tapered capillary of 2.0 µm tip I.D. (**b**) Regression analysis of the number of CCRF-CEM cells released by the SCT device versus the number of cells captured by the SCT device. The x axis indicates the number of captured cells; the y axis is the number of successfully released cells. Error bars show range (n = 4). (**c**) Captured CCRF-CEM cell in the microfluidic device containing herringbone structures. Cells are stained using 4′,6-diamidino-2-phenylindole (DAPI) before spiking to the blood sample. (**d**) Calibration curve between the number of cells spiked into blood samples using SCT and the number of cells captured using the microfluidic device. Error bars show one standard deviation (n = 3).

**Table 1 bioengineering-11-00542-t001:** Comparison of our method with commercial SCT devices.

Project Components	Eppendorf	Olympus	Our Method
Manipulator accuracy	<20 nm	40 nm	150 nm
Assembly components	17	19	7
Price estimate	USD 18,700	USD 18,920	USD 865
Consumables (i.e., capillaries)	USD 595	USD 400	USD 40
Total price (USD)	USD 19,265	USD 19,320	USD 905

## Data Availability

Data are contained within the article and Appendix A.

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
