# Peer review of "A Capillary-Force-Driven, Single-Cell Transfer Method for Studying Rare Cells"

_bioengineering, 2024, doi:10.3390/bioengineering11060542_

Round 1
Reviewer 1 Report
Comments and Suggestions for Authors
I have thoroughly reviewed the manuscript entitled "Capillary force driven, single-cell transfer method for studying rare cells". I think that the current manuscript is an incomplete manuscript for publication because the formation of this manuscript is inadequate and detailed scientific discussion with innovative point is not sufficiently provided. Taking into account the quality of work and scope of the journal, I would recommend the major revision according to the following comments.
Major comments:
# 1. This manuscript investigated the atmospheric pressure driven capillary aspiration. Experimental results of this work are interesting, although the innovative points and scientific discussion of this paper did not demonstrated clearly in the introduction part, results and discussions and conclusion? (discussion?) is not clearly presented.
# 2. How to fix a cost of conventional methods? Micropipette heat-pulling system and microscope are not included in the fixation of cost.
# 3. I think that the accuracy is an important factor of cell aspiration.
# 4. What is the innovation point of this paper? The author should describe the innovation points in the section of introduction and abstract.
# 5. Can this method apply to various types of normal cells including epithelial cell, endothelial cell, and white blood cell?
# 6. The authors should check the style of reference in this journal.
Comments on the Quality of English LanguageModerate editing of English language required
Author Response
Reviewer 1
I have thoroughly reviewed the manuscript entitled "Capillary force driven, single-cell transfer method for studying rare cells". I think that the current manuscript is an incomplete manuscript for publication because the formation of this manuscript is inadequate and detailed scientific discussion with innovative point is not sufficiently provided. Taking into account the quality of work and scope of the journal, I would recommend the major revision according to the following comments.
Major comments:
# 1. This manuscript investigated the atmospheric pressure driven capillary aspiration. Experimental results of this work are interesting, although the innovative points and scientific discussion of this paper did not demonstrated clearly in the introduction part, results and discussions and conclusion? (discussion?) is not clearly presented.
Thank you for the comment. As mentioned in the introduction, our SCT is innovative in two aspects: 1. It can perform sampling and accurately control sampling volume using capillary force without the requirement of sampling handler. 2. The SCT tip can be easily fabricated using temperature-controlled heat pulling. To empathize with the innovation of this work, we added the following sentences.
“In this work, we report our study on the development of atmospheric pressure driven capillary aspiration to facilitatea capillary force driven SCT for accurate CTC concentrationscell manipulation, without the requirement of a sample handler. We evaluate different parameters in micropipette heat-pulling to create varied taper ge-ometriesThe SCT capillary tip can be easily fabricated with high reproducity using temperature controlled heat pulling. Mounted on a X-Y-Z stage, the SCT can locate target cells with high precision. The SCT can be switched to ‘on’ or ‘off’ state by simply disconnected or connected the rare end of the capillary to a syringe. The sampling volume is accurately controlled by the pressure difference of the ‘on’ and ‘off’ stage of the rare end of the capillary.” (Line 65-72)
# 2. How to fix the cost of conventional methods? Micropipette heat-pulling system and microscope are not included in the fixation of cost.
Thank you for this critical comment. We herein present a concept of sampling handler free sampling. The cost comparison is an estimation, but not fully accurate. While a microscope is used for cell localization, it is not an essential part of the system. Instead, it can be replaced by cheaper methods such as amplification lens. We didn’t put the cost of the microscope for other systems either. As for the heat-pulling system, it is not standardized and essential either. A capillary can simply heat-pulled by a ethanol burner. In terms of commercialization, hundreds of capillaries can be mass produced by one heat-pulling system that can be used to fit hundreds of SCT systems. The cost of the heat pulling system can be minimized in mass production.
# 3. I think that the accuracy is an important factor of cell aspiration.
Thank you for the comment. It’s true that accuracy is an important factor. As for the SCT, the volume of sampling is accurately controlled by the capillary. Please refer to the section “3.3 Determination of Equivalent Pressure Change”.
# 4. What is the innovation point of this paper? The author should describe the innovation points in the section of introduction and abstract.
Thank you for the suggestion. We added sentences to describe the innovation points as mentioned in Comment 1.
# 5. Can this method apply to various types of normal cells including epithelial cell, endothelial cell, and white blood cell?
Thank you for this comment. The current SCT platform focuses on single cell transfer, which mainly target floating cells. Therefore, it can easily target suspension cell such as white blood cells. For epithelial cells and endothelial cells, they can be targeted and transferred if they are in suspension state. However, if they are in adhesion state, it is hard to aspirate them without causing destruction of the cells using the SCT. To address this comment, we added the following sentences in the discussion.
“The current SCT platform focuses on single cell transfer, which mainly target floating cells. Therefore, it can easily target suspension cell such as white blood cells. For epi-thelial cells and endothelial cells, they can be targeted and transferred if they are in suspension state. However, if they are in adhesion state, it is hard to aspirate them without causing destruction of the cells using the SCT. For future studies, this system can be combined with adherent cell release technologies such as optical tweezers to manipulate adherent cells.” (Line 311-317)
# 6. The authors should check the style of reference in this journal.
Thank you for the suggestion. We double-checked the references.
Reviewer 2 Report
Comments and Suggestions for Authors
The authors propose a system for the transfer of single cells using capillary forces. Devices with such functions can be useful for various microbiology and medical applications. However, the proposed system is cumbersome, and the operation of capturing cells and transferring them is not automated, which significantly increases the time required to complete the procedures. What is striking is the rather low level of understanding and description of physical processes and processing of the results obtained.
The article cannot be published.
Some major remarks
1. The authors try to explain physics, but they themselves get confused. For example, they write (Line 176): “Once the capillary was positioned near a tumor cell using the micromanipulator, the tubing would be disconnected from the capillary. A small suction pressure (~220.5 Pa as calculated above) was produced via capillary action to attract the cell to the capillary tip.” However, the change in pressure will not be the same as the authors indicate. The pressure difference that arises when the tube is disconnected, which will suck the liquid into the capillary, will be equal to the excess (above atmospheric) pressure that arose previously in a closed volume. It will depend on the size of the enclosed volume, the shape of the tip of the capillary and the angle of insertion of the capillary into the liquid.
2. Figure 4: How can the error at point (1.1) not be equal to zero? The authors indicate an error of ~0.33. It seems that the given graph is not the result of the measurements taken.
Some minor comments.
1. Line 54: It is completely incomprehensible how a 10% error when placing the number of cells can ambiguously affect the performance of microfluidic devices
2. Line 103 et seq.: The description of the method must be moved to the Methods section.
3. Line 121: “At this step only minimal amount of fluid was driven to the capillary by the capillary force as the pressure in the tubing increased when the air was being compressed.” First, this quantity can be simply measured. Secondly, the difference in the volumes of liquids entered with the tube disconnected or connected to the syringe will be minimal, since the volume of liquid entered is much less than the closed volume of the syringe and tube.
4. Line 165: “surface tension is, 𝛾 = 0.0728 J/m2, contact angle, 𝛽 = 32°30” Either a reference indicating the temperature or your own measurement data is required. In addition, these values will significantly depend on the composition of the liquids used.
5. Line 148: “heated to 60 °C to melt and slice accurately.” If the capillaries are made of borosilicate glass, then how can they melt at all? In addition, this temperature is clearly insufficient to soften the material.
6. Line 182: need a link to (3)
7. Line 265 “atmospheric pressure driven”. What does it mean?
8. The authors use the term “Micromanipulation”, which does not correspond to reality.
9. Line 272: “By reducing the process down to a few micropipettes.” What does it mean?
Author Response
Reviewer 2
The authors propose a system for the transfer of single cells using capillary forces. Devices with such functions can be useful for various microbiology and medical applications. However, the proposed system is cumbersome, and the operation of capturing cells and transferring them is not automated, which significantly increases the time required to complete the procedures. What is striking is the rather low level of understanding and description of physical processes and processing of the results obtained.
The article cannot be published.
Some major remarks
- The authors try to explain physics, but they themselves get confused. For example, they write (Line 176): “Once the capillary was positioned near a tumor cell using the micromanipulator, the tubing would be disconnected from the capillary. A small suction pressure (~220.5 Pa as calculated above) was produced via capillary action to attract the cell to the capillary tip.” However, the change in pressure will not be the same as the authors indicate. The pressure difference that arises when the tube is disconnected, which will suck the liquid into the capillary, will be equal to the excess (above atmospheric) pressure that arose previously in a closed volume. It will depend on the size of the enclosed volume, the shape of the tip of the capillary and the angle of insertion of the capillary into the liquid.
The reviewer raised a good point. We did a rough estimation of the suction pressure omitting the small amount of fluid sucked in the capillary when the rare end of the capillary is connected to the syringe. To address this comment, we added a few derivations of the equation to make the estimation more accurate. Please refer to Line 176-Line 205. We also added Figure S1 to illustrate the sampling process and the dimensions referred in the section.
The reviewer is right that the pressure in the SCT can vary with different enclosed volume, the shape of the tip of the capillary. However, the effective capillary driven pressure as calculated in Equ. (1) is the same for the same type of capillary fabricated with the same temperature controlled heat-pulling system. And the reviewer also realized that the pressure difference exists between the ‘on’ and ‘off’ state of the SCT system. The innovative point of the SCT system here is to make use of this pressure difference, and achieve accurate sampling. For the setup of the SCT system, the size of the enclosed volume (3 mL syringe, the tubing plus the capillary), the shape of the tip (the temperature collected fabrication of the capillary tip), and the angle of insertion (the fixed of capillary tip at 45°) are all precisely controlled. Therefore, the volume of sample withdrawn by the SCT each time of sampling is roughly the same.
- Figure 4: How can the error at point (1.1) not be equal to zero? The authors indicate an error of ~0.33. It seems that the given graph is not the result of the measurements taken.
Thank you for the critical comment. For Figure 4, different numbers of cells (1, 5, 10, or 15) were transferred using the SCT system. The number of cells picked up by the SCT can be accurately monitored with the microscope, and they are presented as the X-axis of Fig. 4d. These cells were used for rare cell sample preparation and processed by a microfluidic chip. The captured cells in the microfluidic chip were then enumerated under the microscope (Y-axis of Fig. 4d). The number of cells recovered in the microfluidic chip may not match the number of cells transferred by the SCT since there can be cell loss during cell release or cell sample preparation. For point (1,1), out of 4 times of one-cell transfer, the cell was detected in the microfluidic chip three times. This is where the error bar came from.
Some minor comments.
- Line 54: It is completely incomprehensible how a 10% error when placing the number of cells can ambiguously affect the performance of microfluidic devices
Thank you for the critical comment. The main point here is to emphasize that serial dilution for rare cell sample preparation is not reliable. We re-iterate the content in the main text. (Line 49-53)
- Line 103 et seq.: The description of the method must be moved to the Methods section.
Thank for the comment. The referred section mainly about the scheme and layout of the system as well as the working principle. It would be better to remain in the result section.
- Line 121: “At this step only minimal amount of fluid was driven to the capillary by the capillary force as the pressure in the tubing increased when the air was being compressed.” First, this quantity can be simply measured. Secondly, the difference in the volumes of liquids entered with the tube disconnected or connected to the syringe will be minimal, since the volume of liquid entered is much less than the closed volume of the syringe and tube.
Thank you for the critical comment. This comment is addressed in major comment 1.
- Line 165: “surface tension is, ? = 0.0728 J/m2, contact angle, ? = 32°30” Either a reference indicating the temperature or your own measurement data is required. In addition, these values will significantly depend on the composition of the liquids used.
Thank you for the critical comment. We added a reference for the surface tension using water to approximate the value of DPBS buffer we are using. The contact angle of the buffer drop was measured in house. The temperature is room temperature (~20 ºC).
- Line 148: “heated to 60 °C to melt and slice accurately.” If the capillaries are made of borosilicate glass, then how can they melt at all? In addition, this temperature is clearly insufficient to soften the material.
We apologize for the mistake. it is the voltage reading (60% max puller power). We adjust how we provide this value. We changed the content in the main text. Please refer Line 151.
- Line 182: need a link to (3)
Thank you for the comment. However, it’s a little ambiguous.
- Line 265 “atmospheric pressure driven”. What does it mean?
Sorry for the confusion. We reworded this part. Please refer to Line 300-302.
- The authors use the term “Micromanipulation”, which does not correspond to reality.
Given the amount of fluid the system sampled and the number of cells the system transferred at an accurate manner. We used the term “micromanipulation” by following those commercial “micromanipulators”.
- Line 272: “By reducing the process down to a few micropipettes.” What does it mean?
Sorry for the confusion. We re-worded the referred sentences. Please refer to Line 307-309.
Reviewer 3 Report
Comments and Suggestions for Authors
The research illustrates a new approach for precisely moving single cells, especially unique cells like circulating cancer cells, by leveraging capillary force. This technique paves the way to separate and examine single cells within diverse groups. By sparking capillary force via rapid pressure changes, a tiny portion of the cell membrane is sucked into the reservoir tip, which makes the harmless transfer of single cells possible. The scientists effectively proved the potency of this tactic in producing correct concentrations of scarce cells varying from 1 to 15 cells/mL. It is about the value of looking at unique cells within complex tissues and single-cell analyses in relation to cellular heterogeneity. Here my suggestions for possible enhancements to this manuscript.
How does the rare cell transferring efficiency and accuracy of a capillary-force-driven method compare with those of other methods that are available? Please, clarify further.
Please include recent citations in your references.
How does the non-destructive nature of single-cell transfer using capillary force contribute to preserving cell integrity and downstream analysis?
What are the limitations on scalability and throughput for handling larger sample sizes in terms of the capillary-force-driven approach?
How does complexity in sample preparation affect practicality and reproducibility, if implemented as a method in lab?
Would it be worth performing comparative studies alongside these additional techniques to confirm this technique’s resilience & trustworthiness?
Author Response
Reviewer 3
The research illustrates a new approach for precisely moving single cells, especially unique cells like circulating cancer cells, by leveraging capillary force. This technique paves the way to separate and examine single cells within diverse groups. By sparking capillary force via rapid pressure changes, a tiny portion of the cell membrane is sucked into the reservoir tip, which makes the harmless transfer of single cells possible. The scientists effectively proved the potency of this tactic in producing correct concentrations of scarce cells varying from 1 to 15 cells/mL. It is about the value of looking at unique cells within complex tissues and single-cell analyses in relation to cellular heterogeneity. Here my suggestions for possible enhancements to this manuscript.
How does the rare cell transferring efficiency and accuracy of a capillary-force-driven method compare with those of other methods that are available? Please, clarify further.
Thank you for the critical comment. The most comparable methods are the commercialized sample handler and single-cell manipulators (Ref. 27-30). However, these systems can manipulate cells with high accuracy, but requires liquid handler or pumping units. The key innovation of our SCT system is the use of capillary force and without the requirement of a external pumping system or liquid handler. To address this comment we added the following sentences.
“The transfer efficiency of the SCT is up to 100% efficiency for small cell numbers which is comparable with commercialized single-cell manipulation systems [27-30]. However, for higher cell numbers, the transfer efficiency of SCT slightly decreases (86.7±5.1% for 15 cells), possibly due to the fact that more times of transport is required for bigger number of cells and cell loss is more likely.” (Line 250-254)
Please include recent citations in your references.
Thank you for the suggestion. More references were added in the introduction.
We added citations and some more discussion in the introduction about other capillary-based single manipulation systems.
Thank you for the suggestion. We added more discussion in the introduction. (Line 59-64)
How does the non-destructive nature of single-cell transfer using capillary force contribute to preserving cell integrity and downstream analysis?
Thank you for the comment. In Section 3.3, we calculated the tension added to the aspirated cell as compared with the rupture stress. The tension on the cell is significantly smaller than the rupture stress. Therefore, the effect of capillary force induced tension on the cell is minimal.
What are the limitations on scalability and throughput for handling larger sample sizes in terms of the capillary-force-driven approach?
The limitation of the current SCT system is the relatively low throughput. As mentioned in Section 3.3, each around of sampling, it only aspirates 26 µL of fluid contains one or a few cells. These may be optimized by adding modules like automated screening and machine learning-based target cell localization.
How does complexity in sample preparation affect practicality and reproducibility, if implemented as a method in lab?
Thank you for the question. As shown in the Method section 3.5 and 3.6. For single-cell manipulation, one droplet (~50-100 uL) with several cells can be used for cell transport. The target cell can be transferred to different secondary locations such as microwells, or blood containing tubes. The initial sample is straightforward. However, it is usable only for suspension cells now. The reproducibility of the system was tested in Section 3.5 and 3.6, which shows high efficiency and high reproducibility. To further address this comment, we added the following sentences.
“The current SCT platform focuses on single cell transfer, which mainly target floating cells. Therefore, it can easily target suspension cell such as white blood cells. For epithelial cells and endothelial cells, they can be targeted and transferred if they are in suspension state. However, if they are in adhesion state, it is hard to aspirate them without causing destruction of the cells using the SCT. For future studies, this system can be combined with adherent cell release technologies such as optical tweezers[39] to manipulate ad-herent cells.” (Line 311-317)
Would it be worth performing comparative studies alongside these additional techniques to confirm this technique’s resilience & trustworthiness?
Great suggestion from the reviewer. It would be great to have a comparative study between the SCT system and other systems. Unfortunately, we don’t have a commercialized system available for this comparative study. However, compared with serial dilution for sample preparation, our SCT system assisted sample preparation method is significantly more accurate. (Ref. 12)
Round 2
Reviewer 1 Report
Comments and Suggestions for Authors
The manuscript was corrected to the sufficient level for Bioengineering.
Comments on the Quality of English Language
Minor editing of English language required
Reviewer 2 Report
Comments and Suggestions for Authors
The authors have done some work that has significantly improved the quality of the manuscript. The article can be published in this form.